# Antitumor, Anti-inflammatory and Antiallergic Effects of *Agaricus blazei* Mushroom Extract and the Related Medicinal Basidiomycetes Mushrooms, *Hericium erinaceus* and *Grifola frondosa*: A Review of Preclinical and Clinical Studies

**DOI:** 10.3390/nu12051339

**Published:** 2020-05-08

**Authors:** Geir Hetland, Jon-Magnus Tangen, Faiza Mahmood, Mohammad Reza Mirlashari, Lise Sofie Haug Nissen-Meyer, Ivo Nentwich, Stig Palm Therkelsen, Geir Erland Tjønnfjord, Egil Johnson

**Affiliations:** 1Department of Immunology and Transfusion Medicine, Oslo University Hospital, 0407 Oslo, Norway; uxmoir@ous-hf.no (M.R.M.); lisoha@ous-hf.no (L.S.H.N.-M.); uxneiv@ous-hf.no (I.N.); 2Institute of Clinical Medicine, University of Oslo, 0318 Oslo, Norway; gtjonnfj@ous-hf.no (G.E.T.); egil.johnson@medisin.uio.no (E.J.); 3National CBRNE Medical Advisory Centre, Oslo University Hospital, 0407 Oslo, Norway;uxjmta@ous-hf.no; 4Department of Immunology and Transfusion Medicine, Akershus University Hospital, 1478 Lørenskog, Norway; faiza.mahmood@ahus.no; 5Department of Surgery, Vestfold Hospital, 3103 Tønsberg, Norway; stherk@siv.no; 6Department of Haematology, Oslo University Hospital, 0424 Oslo, Norway; 7KG Jebsen Centre for B-cell Malignancies, Institute of Clinical Medicine, University of Oslo, 0315 Oslo, Norway; 8Department of Gastrointestinal and Pediatric Surgery, Oslo University Hospital, 0407 Oslo, Norway

**Keywords:** antitumor, anti-inflammatory, antiallergic, *Agaricus blazei*, Hericium erinaceus, Grifola frondosa, mushrooms, clinical studies

## Abstract

Since the 1980s, medicinal effects have been documented in scientific studies with the related *Basidiomycota* mushrooms *Agaricus blazei* Murill (AbM), *Hericium erinaceus* (HE) and *Grifola frondosa* (GF) from Brazilian and Eastern traditional medicine. Special focus has been on their antitumor effects, but the mushrooms’ anti-inflammatory and antiallergic properties have also been investigated. The antitumor mechanisms were either direct tumor attack, e.g., apoptosis and metastatic suppression, or indirect defense, e.g., inhibited tumor neovascularization and T helper cell (Th) 1 immune response. The anti-inflammatory mechanisms were a reduction in proinflammatory cytokines, oxidative stress and changed gut microbiota, and the antiallergic mechanism was amelioration of a skewed Th1/Th2 balance. Since a predominant Th2 milieu is also found in cancer, which quite often is caused by a local chronic inflammation, the three conditions—tumor, inflammation and allergy—seem to be linked. Further mechanisms for HE were increased nerve and beneficial gut microbiota growth, and oxidative stress regulation. The medicinal mushrooms AbM, HE and GF appear to be safe, and can, in fact, increase longevity in animal models, possibly due to reduced tumorigenesis and oxidation. This article reviews preclinical and clinical findings with these mushrooms and the mechanisms behind them.

## 1. Introduction

*Agaricus blazei* Murill (AbM) is an edible mushroom of the *Basidomycota* family, growing freely in the coastal Piedade rain forest area outside of São Paulo, Brazil. It was defined in 1967 by the Belgian botanist Paull Heinemann, who named it after the American mycologist William Murill [1]. AbM has mainly been used by the local population as a food ingredient, but also as a remedy against a wide range of diseases, in particular against infection and cancer [2]. In 1965, spores from AbM were brought to Japan, where artificial cultivation and large-scale production was undertaken. At the same time, AbM became the subject of extensive medical research, showing notable immunomodulating and tumoricidal effects, primarily in vitro [1,2]. The fruiting body of AbM is rich in β-glucans, and the biological effect of AbM has originally been attributed to the immunomodulating effect of this polysaccharide, mainly through activation of the innate immune system [3]. β-glucans are chains of D-glucose linked by β-type glycosidic bonds [4], and the main structure of ß-glucans in AbM is a β- (1–3) linked backbone with (1–6) linked side branches. Their molecular weight varies from a few kDa to several thousand kDa, and their biologic effects vary with their structure, molecular weight, and tertiary conformation [5]. In addition, AbM contains metabolic substances which exhibit cytotoxic effects, such as the steroid blazein, the lipid ergosterol, its derivate agarol, and the fenolhydrazine containing compound agaritine [6,7,8]. Although several mushrooms in the Basidiomycetes family have similar functions to those examined here, the related *Agaricus blazei* Murill (AbM), *Hericium erinaceus* (HE) and *Grifola frondosa* (GF) mushrooms were focused on in this review because they are components of Andosan^TM^, which has been extensively investigated in clinical studies.

HE and GF are also rich in β-glucans, and they have both been shown to have immunomodulating and antitumor effects [9,10]. HE (“lion’s mane”) and GF (“hen of the woods”) grow on hardwood trees in America and Asia and are both widely consumed, especially in the Far East, because of their nutritional qualities and perceived health benefits. In addition to glycoproteins and polysaccharides, HE contains a number of metabolic substances, in particular the aromatic compounds hericerins and erinacines, which have been shown to have a function as a nerve growth factor [11]. Research on GF has particularly focused on a proteoglycan, called the D-fraction, which has been shown to have immunomodulatory, cytotoxic and antioxidant effects in vitro [12]. HE and GF are minor ingredients (15% and 3%, respectively) in the AbM-based mushroom product Andosan^TM^ (ACE Co. Ltd. produced for Immunopharma, Gifu-ken, Japan) which first was shown to have significant anti-infective effects in mouse models of bacterial sepsis [13,14], and later to have antiallergic [15] and antitumor effects [16] as well.

Chronic inflammatory conditions per se or infections giving rise to a chronic inflammatory response tend to predispose cancer development. Examples are: hepatitis B and C infection and hepatocellular carcinoma [17], pancreatitis and pancreatic cancer [18], atrophic gastritis and gastric cancer [19], *Helicobacter pylori* infection and mucosa-associated lymphoid tissue (MALT) lymphoma, which usually regress after antibiotic treatment [20], bladder infection with *Schistosomia hematobium* and bladder cancer [21], prostatitis and prostatic cancer [22], and inflammatory bowel disease and colorectal cancer [23,24]. Hence, there is a linkage between inflammation and tumorigenesis.

An increased T helper cell (Th) 2 response promotes allergy and asthma development. Th2 is also dominant in the cancer environment [25], where a T helper cell (Th) 1 immune response is essential for antitumor activity [26]. Since the Th1/Th2 balance is inversely linked [27], one may speculate whether there is a link between allergy/asthma and cancer. This seems to be controversial; while one study found increased risk of colorectal cancer in allergic disease [28], another study did not find any association between allergy and risk for breast or prostate cancer [29]. Moreover, since allergy and asthma can be regarded as topic inflammation induced by different allergens, e.g., in the eyes and nose for pollen, in the throat or skin for food components, and in the lungs for inhalants, these atopic conditions could also predispose for cancer. In fact, a recent meta-analysis could not rule out a possible association between atopy and malignancy of lung, skin and oesophagus [30]. The hygiene hypothesis has also established a link between protection by early presence of bacterial and parasitic infections against allergy and asthma development in childhood [31]. An interesting twist in this respect is our previous finding of increased allergy among tuberculosis (TB) and leprosy patients relative to their healthy controls [32,33], which was explained by a reduced defense against TB infection, i.e., low Th1 response, in approximately 10% of individuals, who normally contract TB after exposure to *M. tuberculosis* [33].

The aim of this paper was therefore to review preclinical and clinical studies on the antitumor, anti-inflammatory, and antiallergic effects of AbM and its related medicinal mushrooms GF and HE. Selection criteria were inclusion of PubMed/Medline indexed articles on the above in vivo results with these Basidiomycetes mushrooms, and exclusion of articles on findings in other areas.

## 2. Material and Methods

### 2.1. Antitumor Effects of ABM-Preclinical Studies

Since the early 1990s, there has been a host of studies on preclinical antitumor effects of AbM extracts in rodent models (Table 1). At first, fibrosarcoma was studied in a double grafted mouse model, and both a polysaccharide–protein complex and lipid fraction of AbM were found to inhibit tumor growth after oral and intraperitoneal administration [34,35,36]. This was thought to be due to immunological- and ergosterol-mediated inhibited neovascularization of tumor. However, Delmanto et al. [37] suggested that the mechanism for the observed antitumor effect of AbM was the mushroom’s antimutagenic effect, as demonstrated by its reduction in cyclophosphamide-induced micronuclei in the bone marrow and blood cells of the treated mice. The antitumor and antimetastatic effect of AbM polysaccharide in the fibrosarcoma model was confirmed after intratumor injection in 2005 [38], oral administration of AbM hot water extract in 2016 [39], and fermented AbM mycelia in 2018 [40]. The mechanism in the latter study was suggested to be an increase in cluster of differentiation (CD) 4+ and CD8+ T cells and a reduction in CD19+ B cells [40], restoring the balance between cellular and humoral immunity.

One study also found that AbM extract, combined with marine phospholipid to increase its uptake, suppressed growth of myeloma cells in a mouse model [50]. Another study reported the inhibition of prostate cancer in a mouse model by a β-glucan-enriched AbM froth fraction by means of apoptosis and antiangiogenesis [49]. Niu et al. [47,48] performed extensive studies in mouse models for melanoma and sarcoma and showed that a low molecular weight (LMW) AbM polysaccharide could reduce lung metastasis of the former and growth and metastasis of the latter. Mechanisms for reduced tumor growth were an antiangiogenetic effect [47], and for the antimetastatic effect, modulation was mediated by metalloproteinase-9 and nm23-H1 [48], a metastatic suppressor [51]. On the other hand, Ziolotto et al. found no effect on colon carcinogenesis of AbM given orally to Wistar rats [46]. This is, however, in contrast to findings referred to below. Wu et al. [43,44] showed that AbM extract inhibited growth of colon cancer, hepatoma and melanoma in severe combined immunodeficient (SCID) mice and increased their life span in a dose-dependent manner. The finding of AbM (combined with chitosan N-acetyl glucosamine) inhibiting hepatoma cell growth in SCID mice was confirmed later [41]. Pinto et al. [45] observed that a β-glucan-rich AbM preparation reduced growth of Ehrlich tumor after intratumor injection, increased blood levels of IFNγ, CD4+ T cells and macrophages (MΦ), and reduced interleukin (IL)-10, resulting in immune cell migration to tumor and cytokine switch.

In a study of murine leukemia, an AbM extract reduced liver and spleen sizes, increased interferon (IFN)γ, IL-1β, IL-6 and reduced IL-4 levels [42]. The authors related this to increased CD3+ and CD19+ cells and reduced MΦ. Moreover, ex vivo experiments demonstrated that the Andosan^TM^ extract (containing mainly AbM, see above) had a cytotoxic effect on primary myeloma cells, and also on myeloma and leukemia cell lines in vitro, probably caused by cell cycle arrest [52]. One study found an anticancer effect of an ergosterol derivative (Agarol) from AbM in a xenografted carcinoma murine model due to apoptosis [7]. We have reported that the addition of Andosan^TM^ to drinking water protected mice against tumorigenesis of intestinal adenocarcinoma, which develops spontaneously from polyposis in A/J Min mice with a deletion in the *apc* gene [16]. The mechanism was immunomodulatory, as shown by increased IL-12 levels, and the growth inhibition of tumor cells by the induction of apoptosis. Recently, Sovrani et al. [53] reported that AbM exobiopolymers inhibited solid Walker 256 tumors in rats due to increased nitric oxide (NO^-^) production by peritoneal MΦ.

In summary, the mechanisms for the AbM-induced antitumor effects in rodent models were the following: inhibition of tumor neovascularization [35,36,47,49], antimutagenesis [37], modulation of metalloproteinase and a metastatic suppressor [48], Th1 immune cell migration to tumor [45], apoptosis [7,16,49,52], and increased NO^-^ production by MΦ [53].

### 2.2. Antitumor Effects of AbM—Clinical Studies

In 1994, it was reported that treatment with AbM had an inhibitory effect on malignant cells in patients with acute non-lymphoblastic leukaemia [54] (Table 2). Ten years later, Ahn et al. [55] treated 100 patients with gynecological cancer who received chemotherapy with add-on AbM extract, and found that this, in contrast to add-on placebo, increased natural killer (NK) cell activity and improved quality of life (QoL) [55].

In a pilot study with Andosan^TM^, in the treatment of five patients with chronic hepatitis C virus (HCV) infection for a week, a slight, but statistically non-significant, reduction in serum HCV load was observed [58]. In addition, there was a surprising finding: microarray analysis of their peripheral leukocytes showed an increased expression of genes inducing apoptosis and inhibition of cell division, which is related to tumor defense [58]. In a recent placebo-controlled and randomized clinical study by Tangen et al. [56], Andosan^TM^ extract was given orally for seven weeks as adjuvant treatment to half of 40 multiple myeloma patients undergoing high-dose chemotherapy with autologous stem cell transplantation (ASCT). The immunomodulatory findings were an increased number of plasmacytoid dendritic cells (pDC) and T regulatory cells (T_reg_s) in the blood, increased serum levels of IL-1**ra** (**r**eceptor **a**ntagonist), IL-5 and IL-7, and enhanced expression of immunoglobulin genes, Killer Immunoglobulin Receptor (KIR) genes and human leukocyte (HLA) genes in the bone marrow in the Andosan^TM^ group. Although time with intravenous (i.v.) antibiotics during aplasia was 1.5 days shorter, and time to second line treatment and overall survival (OS) was 6 and 3.3 months longer, respectively, no statistically significant clinical impact of Andosan^TM^ was detected at follow-up after 4 years. At a second follow-up after 5.7 years, mean OS was 79.0 months (95% CI: 61.5–96.5 months) in the Andosan^TM^ group against 65.8 months (95% confidence interval (CI): 50.5–80.9 months) in the placebo group (mean observation time = 67.7 months) (*p* = 0.16)(Figure 1). Moreover, mean time to second line treatment was 47.3 months (95% CI: 30.5–64.0 months) in the Andosan^TM^ group and 38.0 months (95% CI: 25.1–50.9 months) (median observation time 3.5 years)(*p* = 0.39) (unpublished data).

There is one clinical study on prostate cancer with an AbM extract (Senseiro), which found prolonged doubling time of prostate specific antigen (PSA) levels. However, since it was not correlated with testosterone production and placebo controls were lacking, the authors concluded with no significant effect of AbM [57].

### 2.3. Antitumor Effects of GF

Studies of the anti-tumor effects of GF polysaccharides began in 1985, when the repression of implanted Meth A sarcoma was found in a mouse model. This was due to the stimulation of adaptive immunity, i.e., an increase in spleen cells [59] and cytotoxic T cells, as well as MΦ [60] (Table 3). A GF (fraction D) β-glucan had an antitumor effect on colon carcinoma implanted in mice, which was related to the establishment of Th1 dominance in a population that was Th2-dominant due to carcinoma [25]. A year later, this GF β-glucan was tested in 10 cancer patients (with lung, lingual, breast, gastric, or liver cancer) and reported to inhibit the progression of metastasis and reduce the expression of tumor markers (carcinoembryonic antigen (CEA), cancer antigen (CA)15-3, and CA19-9). This was primarily caused by an increase in NK cell activity and Th1 response and, inversely, a reduction in Th2 activity [61]. The same group at Kobe Pharmaceutical University, Japan, confirmed that the GF β-glucan could reduce colon cancer growth in mice by inducing cell-mediated immunity and Th1 response [62,63]. They further showed that GF β-glucan both enhanced antitumor and antimetastatic effects of cisplatin and reduced nephrotoxicity [64].

A LMW protein fraction of GF inhibited implanted colon carcinoma in mice, which was thought to be due both to increased IL-1β, TNFα, and, surprisingly, IL-10. Furthermore, an increase was found in the Th1 cytokines IL-12 and IFNγ, and in activated MΦ, NK and dendritic cells [68]. A systemic antitumor response of the GF β-glucan in mice was caused by immunomodulation, including the activation of MΦ in Peyer’s patches and an increase in IFNγ [63]. Moreover, the antitumor effect of a GF extract on implanted kidney cancer has been shown in rats, due to immunomodulation and tumor necrosis [67]. Recently, a selenium-enriched GF polysaccharide has been reported to have an enhanced antitumor effect in hepatoma (hepatocellular carcinoma (*Heps*))-bearing mice by improved immune function [66]. Since the GF β-glucan also enhanced the antitumor activity of 5-Fluorouracil (5-FU) in addition to protecting against its side effects, GF β-glucan could be developed as an auxiliary substance for chemotherapeutic drugs [65].

To summarize the mechanisms behind the GF-induced antitumor effects, they were: increase in spleen cells, Th1 and cytotoxic T cells, activated MΦ (also in Peyers patches) and NK cell activity [25,59,60,62,64], enhanced effects and reduced side effects of cisplatin and 5-FU [63,65], and necrosis [67].

### 2.4. Antitumor Effects of HE

In a murine model with a xenografted human colon cancer cell line, Kim et al. [9] found that aqueous and aqueous/ethanol extracts of HE [69] injected intraperitoneally led to the regression of the tumors. This effect was associated with an increase in intraperitoneal MΦ and serum pro-inflammatory cytokines. Furthermore, increased expression of genes coding for vascular endothelial growth factor (VEGF), cyclooxygenase 2 (Cox 2) and 5-lipooxygenase (5-LOX) were noted [9]. It was shown that treatment with these extracts also inhibited migration of the cancer cells to the lungs by 66% and 69%, respectively [70]. In these animals, a reduced expression was found of the matrix metalloproteinases MMP-2 and MMP-9 in the cancer cells, possibly causing inhibition of migration and invasion. In another murine model with xenografted human liver cancer-, gastric cancer- and colon cancer cells Li et al. [71] found that two extracts from HE, called HTJ5 and HTJ5A, with a mixed content of aromatic compounds, dipeptides indoles and amino acids, also exhibited a considerable antitumor activity against all cell lines.

## 3. Anti-inflammatory Effects of AbM and HE

### 3.1. AbM

Animal data (Table 4).

AbM extract given orally has been found to reduce carcinogen-induced lung damage in rats due to the attenuation of pulmonary inflammation [74]. Interestingly, in a murine malaria model, it was shown that AbM extract counteracted the deteriorating consequence of cerebral malaria by reducing proinflammatory cytokines TNFα, IL-6 and IL-1β [73]. Very recently, another AbM extract was found to prevent non-alcoholic steatosis by reducing hepatic stress in mice [72].

Human data (Table 5).

In contrast to the proinflammatory effects of Andosan^TM^ found on human monocytic and endothelial cells cultured in vitro and on whole blood cells ex vivo [87,88], a significant *decrease* in IL-1β, TNFα, IL-17 and IL-2 was observed in ten volunteers who ingested Andosan^TM^ for 12 days, showing a predominantly anti-inflammatory effect in vivo [87] (Table 3). Furthermore, in eight healthy volunteers who took Andosan^TM^, increased preshedding expression of the adhesion molecules CD62L (L-selectin) was found, while expression of the adhesion molecules CD11b and CD11c remained unchanged and intracellular reactive oxygen species (iROS) decreased both in monocytes and granulocytes [86].

Twenty-one patients with ulcerative colitis (UC) (*n* = 10) and Crohn’s disease (CD) (*n* = 11) received Andosan^TM^ orally, alone or in addition to standard medical treatment, for 12 days. In both groups, plasma levels of several proinflammatory cytokines and chemokines in lipopolysaccharide (LPS) stimulated blood ex vivo were reduced, as well as fecal content of the pro-inflammatory marker, calprotectin, in the UC patients [85]. In another clinical trial 50 patients with UC and 50 patients with CD were randomized to receive either 60 mL of Andosan^TM^ or placebo for 21 days. Plasma levels of IL-5 (UC group) and IL-2 (CD group) were reduced in patients receiving Andosan^TM^ [82], and patients in both groups enjoyed an improvement in clinical symptoms and quality of life [83,84]. When looking at IL-1ß, IL-6 and granulocyte-colony stimulating factor (G-CSF) combined in the patients with CD, the cytokine levels were significantly lower in the Andosan^TM^ group. In addition, total fatigue was improved in both the patients with UC and CD. The difference found in the in vitro and in vivo effects of Andosan^TM^ is intriguing, and the reason for this discrepancy has not been clarified. However, potentially absorbable LMW substances like flavonoids and other less defined substances [89,90,91] may contribute to the anti-inflammatory effect of AbM in vivo.

### 3.2. HE

Anti-inflammatory properties have also been attributed to HE (Table 4). This is shown for HE itself, and also for erinacine A from HE, both of which protected against brain-ischemia-induced neuronal cell death in rats [78]. The mechanism was the inhibition of inducible nitric oxide synthase (iNOS) and mitogen-activated protein kinases (MAPK), reduced proinflammatory cytokines and nerve growth properties of the mushroom [78]. Another study in rats found that the HE β-glucan did improve inflammatory bowel disease (IBD)-induced colonic mucosa changes, because it promoted the growth of beneficial gut bacteria, which may have improved host immunity by reducing the activation of myeloperoxidase (MPO), nuclear factor kappa B (NF_K_B) and T cells [77]. Moreover, a HE polysaccharide reportedly attenuated colitis in mice by regulating oxidative stress through inflammation-related signaling pathways, composition of gut microbiota, and maintenance of an intestinal barrier [76]. Most interestingly, a HE fraction has been found to increase longevity in aged and carcinoma-bearing mice by the induction of endogenous antioxidative enzymes [75].

In summary, the mechanisms behind the anti-inflammatory effects induced in vivo by AbM were: predominant decrease in proinflammatory cytokines in healthy individuals and IBD patients [82,87], increased shedding of the adhesion molecule leukocyte-selectin and reduced iROS [86]. Anti-inflammatory mechanisms induced by HE were: educed proinflammatory cytokines, inhibition of iNOS, increased nerve growth protecting against neuron death in brain ischemia [78], growth of beneficial gut microbiota protecting against IBD-induced mucosa damages and improving host immunity [77], regulation of oxidative stress through signaling pathways attenuating colitis [76], and endogenous antioxidative enzymes increasing longevity [75].

### 3.3. GF

Regarding GF, it has been shown that a polysaccharide given orally had a similar protective effect on non-alcoholic steato-hepatitis in rats as AbM had in mice above [66]. The mechanism was a beneficial regulation of gut microbiota [79]. Moreover, a fermented GF extract had anti-inflammatory effect in an endotoxin-induced uveitis model in rats [80], and a GF extract reduced colon ulceration in an IBD rat model through an antioxidative and anti-inflammatory mechanism [81].

## 4. Antiallergic Effects of AbM and GF Extracts

It was reported in 2006 [92] that an AbM extract inhibited induced anaphylactic reaction (and also passive immunization), i.e., ear-swelling, in a mouse model by means of a treatment effect on the mast cell reaction (Table 6). In another mouse model for induction of allergic asthma, an AbM extract given orally, reduced levels of specific immunoglobulin (Ig)E, IgG1 and bronchial eosinophils due to the amelioration of skewed Th1/Th2 balance [93]. The finding was confirmed the year after by us with Andosan^TM^ in the similar ovalbumin (OVA)-induced allergic sensitization mouse model, where specific IgE and IgG1 were also reduced and Th1 response increased relative to Th2 response [15]. When this established OVA sensitization model for allergy in the mouse was employed again, it was found that the mechanism behind the reduced specific IgE and improved Th1/Th2 balance was MΦ cell activation by epithelial cells and AbM promotion of differentiation of naïve T cells to Th1 cells [94]. Recently, our group performed a placebo-controlled randomized clinical study in which blood donors with self-reported and specific IgE-confirmed birch allergy and asthma, given Andosan^TM^ orally for 2 months before the pollen season, had less general allergy and asthma ailments and used less medication. This was due to reduced specific IgE levels and reduced mast cell sensitization, as demonstrated indirectly by the basophil activation test [95].

Antiallergic effects have also been observed in mice after the oral administration of a GF polysaccharide or extract (Table 6), where atopic dermatitis-like skin lesions and mast cell degranulation were inhibited, respectively, owing to alleviated anaphylactic cutaneous response [96,97]. The authors found that this was caused by reduced IgE and mast cell infiltration, a cytokine expression ameliorating the Th1/Th2 imbalance [97], and a reduced type I allergic response by suppression of mast cell degranulation [96]. Accordingly, GF polysaccharides could be used as a novel therapeutic agent, replacing corticosteroids, or as a supplementary substance [97].

## 5. Safety of AbM, GF and HE

Three cases of severe hepatic dysfunction have been reported in cancer patients using an oral AbM extract [98] (Table 7). However, other causes of hepatic dysfunction could not be ruled out in these patients. Furthermore, a case of allergic chronic cheilitis possibly occurred following daily AbM intake for 6 months [99]. AbM has been found to inhibit cytochrome P-450 and the trans-membrane-efflux pump P-glycoprotein (P-gp) to a mild degree in vitro (similar to green tea) [100]. AbM should therefore, as a precaution, not be used together with drugs that are P-gp substrates, such as vinblastine, vincristine, digitoxin, cyclosporine, loperamide, verapamil, quinidine and others [101]. One report found negative genotoxicity tests with AbM extracts in rats [102]. In a toxicity study in rats ingesting AbM over 2 years, there was no carcinogenicity or other adverse health effects of AbM. Rather, significantly lower mortality was found among the male rats on AbM treatment [103]. There is a case report of occupational hypersensitivity pneumonitis to GF spore after work in a mushroom farm [104]. Interestingly, HE protein fraction has been found to increase longevity in aged mice by means of antioxidative mechanism [76].

## 6. Discussion

Here, we review preclinical and clinical studies with the three related medicinal mushrooms AbM, HE and GF, where all exhibited both antitumor, anti-inflammatory and antiallergic properties, except for HE that did not reveal antiallergic effects. The antitumor mechanisms were either direct tumor attack, i.e., apoptosis/necrosis, antimutagenesis, and metastatic suppression, or indirect defence, i.e., inhibited neovascularization, Th1 cytotoxic cell tumor migration and increased MΦ NO production, NK cell activation, and enhanced effects and reduced side effects of chemotherapeutic drugs. The anti-inflammatory mechanisms were decreased in proinflammatory cytokines, adhesion molecules, iROS and iNOS. Furthermore, for HE, increased nerve and beneficial gut microbiota growth, and oxidative stress regulation, were noted. The antiallergic mechanism was amelioration of the skewed Th1/Th2 balance found in allergy and asthma. Regarding safety, besides allergic reactions (AbM and GF), undocumented hepatotoxicity (AbM) in case reports, and advised caution with simultaneous use of drugs affecting the intestinal trans-membrane-efflux pump (AbM), these medicinal mushrooms appear to be safe and can, in fact, increase longevity in animal models by reducing tumorigenesis (AbM) or an antioxidative mechanism (HE).

Most animal studies with AbM have been performed with grounded powder or extracts of the fruiting body with high β-glucan contents, and probably with varying batch-to-batch quality. In early studies [38,59], tumor growth suppression by mushroom fruiting body extracts or isolated β-glucan was established by direct intratumor injections. However, it is well-documented that β-glucan is taken up actively from intestines of rodents and brought to the reticulo-endothelial (RE) system [106]. This may explain the proinflammatory and immunomodulating effects observed with AbM in animal models referred to above, including with Andosan^TM^ [16] where IL-1β, monocyte chemoattractant protein-1 (MCP-1) and tumor necrosis factor α (TNFα) were increased in addition to IL-12. This is opposite to the anti-inflammatory effect of Andosan^TM^ found in healthy individuals and IBD patients [82,87]. In humans, uptake of β-glucan, which are often large cellulose polysaccharides [106], has not been documented, but there are studies arguing that it occurs [107,108]. Although Andosan^TM^ has been found to contain only small amounts of β-glucan [109], it has a clear proinflammatory effect in vitro [87,88]. Most probably, this is caused by specific binding to receptors on immune cells (i.e., toll-like receptor 2 (TLR2), dectin-1, complement receptor 3 (CR3)) [106] of β-glucan still present in this extract, which then must have a high affinity to these receptors. Andosan^TM^, which is made from the mushroom mycelium, is one of the commercial AbM products that has been most investigated and shows promise in clinical studies. The mycelium may contain other or more effective ingredients than the fruiting body.

In vivo, there are other LMW molecules in AbM and other mechanisms involved than those involved in vitro. Such LMW components of AbM with anti-inflammatory and antioxidative effects, are isoflavonoids and components from aqueous and alkaline extracts that may be absorbed through the intestinal mucosa in order to exert their effects [89,90,91]. This agrees with our previous observation that the antiallergic effect of Andosan^TM^ in the mouse allergy model seemed to be due to LMW substances [15]. In cultures, the cells are exposed to the whole mushroom extract, in which β-glucans probably drive the proinflammatory response. The different results on intestinal tumorigenesis found in rodents by use of AbM fruiting body (no effect) [46] versus with an AbM-based mycelium extract (Andosan^TM^) (60% reduced tumor load), may be due to the different AbM preparations used.

The different effects of the various AbM extracts can be due to differences in AbM subspecies, the production process, including type of extraction method, and whether the fruiting body was used or the mycelia, as mentioned. This was demonstrated in a blinded comparison of five AbM extracts from major Japanese health food producers in the pneumococcal sepsis model in mice [110]. When given orally before bacterial challenge i.p., only one of the extracts, later called Andosan^TM^, significantly reduced bacteremia and increased survival rate as compared with the saline control. Hence, the different mushroom extracts, even from the same subspecies, may contain different constituents or the required effective concentrations thereof. Moreover, Andosan^TM^ contains extracts and substances from three different mushrooms, AbM, HE and GF, which also may have a positive synergistic effect, but this is unknown.

In the randomized clinical trial with multiple myeloma patients, there seems to be a positive effect, with 1.3 years longer survival in the Andosan^TM^-treated patients. This suggestion should be addressed in a study that is properly powered.

In the A/J Min mice model for colorectal cancer, animals treated with Andosan^TM^ had reduced intestinal tumor load, and also expressed less of the tumor-associated protease, legumain, in their intestines [16]. The fact that this enzyme is also pro-inflammatory in nature, demonstrates a link between inflammation and tumorigenesis. Moreover, Andosan^TM^ also had a systemic proinflammatory effect in these mice, as shown by increased levels of proinflammatory cytokines, which is similar to the mushroom-extract-treated mice in a sepsis model [13]. Hence, Andosan^TM^ has a general proinflammatory effect in mice, in contrast to humans, which is in agreement with the active intestinal uptake of β-glucan in mice [106]. However, this also varies in humans because, in cancer patients given increasing doses of a GF polysaccharide orally, there was a non-monotonic and fluctuating association with both immune-stimulatory and –inhibitory systemic effects [105]. In humans, the main results found with Andosan^TM^ probably is due to β-glucan stimulation of Peyer’s patches in the gut-associated-lymphoid–tissue (GALT) [111], together with other less defined absorbable LMW substances such as flavonoids. However, the main active principle in Andosan^TM^ still remains unknown.

Interestingly, AbM also had antitumor effect on implanted tumor in the very same animals where the antiallergic and antiasthmatic effect of the mushroom extract was shown [92]. At least from an immunological perspective, this illustrates a possible relationship between cancer and allergy.

## 7. Conclusions

This review on scientific in vivo findings with three of the most well-known medicinal mushrooms, AbM, HE and GF, shows that they possess valuable antitumor, anti-inflammatory and antiallergic properties. These should be investigated more thoroughly with quality-controlled and standardized mushroom extracts, or fractions thereof, in randomized clinical studies powered to show putative therapeutic effects. In this way, medicinal *Basidomycota* could become significant supplementary drugs in Western medicine.

## Figures and Tables

**Figure 1 nutrients-12-01339-f001:**
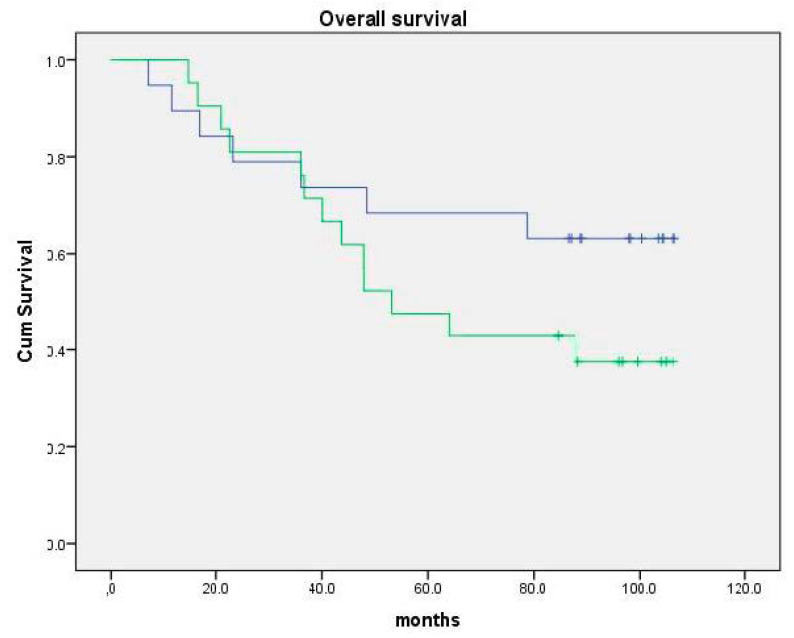
Follow-up of overall survival (OS) of patients included in the clinical study: Andosan^TM^ as adjuvant treatment in MM patients receiving autologous stem cell transplantation (ASCT) [56]. Upper curve: Andosan^TM^ group (*n* = 19). Median OS = 79.0 months (95% CI 61.5–96.5 months). Lower curve: Placebo group (*n* = 21). Median OS = 65.8 months (95% CI 50.5–80.9 months) Median observation time 67.7 months. *p* = 0.16. Previously unpublished result.

**Table 1 nutrients-12-01339-t001:** Antitumor effects of *Agaricus blazei* Murill—Preclinical studies.

Product Admin.	Study In, Of	Effects	Mechanism	Author, Year [Ref]
Solid-state fermented mycelia, p.o.	Mice, Sarcoma	Growth inhibition	Immunomodulation ↑ T cell and ↓ B cell	Rubel et al., 2018 [40]
Ergosterol deriv. (Agarol), i.p.	SCID Mice, Lung Adenoarcinoma	Inhibition	Apoptosis	Shimizu et al., 2016 [7]
Water mycelia extract incl. HE, GF, p.o. (Andosan)	Min Mice, spon. Adenocarcinoma	Inhibition	Immunomodulation, Apoptosis	Hetland et al., 2016 [16]
Water mycelia extract, p.o.	Mice, Sarcoma	Antitumor	Immunoprotection	Bertéli et al., 2016 [39]
Water extract + Chitosan, p.o.	SCID Mice, Hepatoma	Inhibition	Angiogenetic effect	Yeh et al., 2015 [41]
Water extract, p.o.	Mice, Murine Leukemia	↓ Liver & spleen size ↑ IL-1β,IL-6, IFNγ, ↓ IL-4	↑ T% B cells, ↓ MΦ Immunomodulation	Lin et al., 2012 [42]
Water extract, p.o.	SCID Mice, colon cancer, Hepatoma Melanoma	Inhibition, Inhibition, ↑ Life span	Dose-dependent Immunomodulation	Wu et al., 2011 [43,44]
β-glucan-rich extract, i.t.	Mice, Ehrlich, tumor	Inhibition, ↑ IFNγ, T cc, MΦ, ↓ IL-10	Immune cell tumor migration Cytokine switch, Apoptosis	Pinto et al., 2009 [45]
Crude fruiting body, p.o.	Rats, colon ca.	No effect on colon carcinogenesis	-	Ziliotto et al., 2009 [46]
LMW polysacc., i.t., i.p.	Mice, Sarcoma Melanoma	Inhibition ↓ Lung metastasis	Inhib. Angiogenesis Metalloproteinase modulation	Niu et al., 2009 [47,48]
Broth fraction, p.o.	SCID mice, Prostate ca.	Inhibition	Apoptosis, Antiangiogenesis	Yu et al., 2009 [49]
Water extract + marine phospholipid, p.o.	Nude Mice, Myeloma	Inhibition	Immunomodulation ↑ uptake of extract	Murakawa et al., 2007 [50]
Ergosterol, p.o.	Mice, Sarcoma	Inhibition	↓ Neo-vascularization	Takaku et al., 2001 [36]
Water extract, p.o.	Mice, CP-induced clastogenicity	Anticarcinogenic	Antimutagenic	Delmanto et al., 2001 [37]
Mycelia polysaccprotein complex, i.p., p.o.	Mice, Sarcoma	Inhibition	Immunological	Ito et al., 1997 [35]

Abbreviations: Severe combined immunodeficiency (SCID), intraperitoneally (i.p.), perorally (p.o.), intratumor (i.t.), interleukin (IL), interferon (IFN), macrophages (MΦ), cancer (ca.), cells (cc), low molecular weight (LMW), cyclophosphamide (CP).

**Table 2 nutrients-12-01339-t002:** *Antitumor effects of* Agaricus blazei *Murill–Clinical studies*.

Product Admin.	Study In, Of	Effects	Mechanism	Author, Year [Ref]
Water mycelia extract incl. HE, GF, p.o. (Andosan)	Placebo-ctr RCT Myeloma pat. (*n* = 40)	Immunomodula-ting	Immunomodulation	Tangen et al., 2015 [56]
Water extract, p.o. (Senseiro)	Prostate ca. patients (*n* = 32)	Longer PSA doubling time, no testosterone correlation	-	Yoshimura et al., 2010 [57]
Water mycelia extract incl. HE, GF, p.o. (Andosan)	Patient with chron. HCV infection (*n* = 5)	↑ Expression of “antitumor” genes	Immunomodulation	Grinde et al., 2006 [58]
Water extract, p.o.	Gynecol. ca. patients (*n* = 100)	↑ QoL, ↑ NK cc activ.	Immunomodulation	Ahn et al., 2004 [55]
Water extract, p.o.	Ac. non-lymphoblastic leukemic patients (*n* = 10)	Inhibition	-	Hui et al., 1994 [54]

Abbreviations: *Hericium erinaceus* (HE), *Grifola frondosa* (GF), controls (ctr), randomized clinical trial (RCT), patients (pat.), prostate specific antigen (PSA), hepatitis C virus (HSV), quality of life (QoL), natural killer (NK).

**Table 3 nutrients-12-01339-t003:** Antitumor effect of *Grifola fondosa*
(Human study).

Product, Admin.	Study In, Of	Effects	Mechanism	Author, Year [Ref]
GF β-glucan, p.o.	Mice, Heps tumor (hepatoma)	↑ Inhibitory effect of 5-Fu against Heps-tumor	Synergism and immune regulation	Mao et al., 2019 [65]
Selenium-enriched GF polysaccharide, p.o.	Mice, Heps tumor (hepatoma)	Inhibition of Heps-tumor	Improved immune function	Mao et al., 2018 [66]
GF β-glucan, p.o.	Mice, Colon tumor, Melanoma	Inhibition, Systemic antitumor response ↑ Survival	MΦ activ. in Peyer’s patches and ↑ IFNγ	Masuda et al., 2017 [63]
GF mycelia extract, p.o.	Rats, Kidney ca.	Inhibition	Immunomodulation and tumor necrosis	Vetchinkina et al., 2016 [67]
GF LMW protein fraction, i.p.	Mice, Colon ca.	Inhibition	↑ IL-1, TNFα, IL-10, IL-12, IFNγ, activ. NK and DC cc, MΦ	Kodama et al., 2010 [68]
GF polysacc., i.p.	Mice, Colon ca.	Inhibition	Induced cell mediated immunity ↑ Th1 cytokines	Masuda et al., 2009 [62]
GF polysacc., i.p.	Mice, Colon ca.	↑ Antitumor and-metastatic effect on cisplatin, ↓myelo- and nephrotoxicity	Synergistic effect of cisplatin cytotoxicity and GF immunomodulation	Masuda et al., 2009 [64]
GF D fraction, p.o.	Diff. Cancer pat. (*n* = 10)	Hindered metastat. prog. ↓expression of tumor markers	Immunomodulation ↑NK cell activ. and ↑Th 1/↓Th2	Kodama et al., 2003 [61]
GF β-glucan, i.p.	Mice, Colon ca.	Inhibition	Induced cellular mediated immunity and Th1 dominance	Kodama et al., 2002 [25]
GF β-glucan, i.p.	Mice, Sarcoma, Carcinoma	Inhibition	Host-mediated mechanism involving MΦ and cytotoxic T cc	Takeyama et al., 1987 [60]
GF polysacc. fraction, i.p., i.t.	Mice, Sarcoma	Repression	↑ Weight of spleen cc and number	Suzuki et al., 1985 [59]

Abbreviations: hepatocellular carcinoma (Heps), 5-Fluorouracil (5-Fu).

**Table 4 nutrients-12-01339-t004:** Anti-inflammatory effect of *Agaricus blazei* Murill, *Hericium erinaceus* and *Grifola frondosa*-Preclinical studies.

Product, Admin.	Study In, Of	Effects	Mechanism	Author, Year [Ref]
AbM dry feed, p.o.	Mice, non-alcoholic steato-hepatitis	Prevention	Prevention of oxidative stress	Nakamura et al., 2019 [72]
AbM water extract fractions	Mice, cerebral malaria	Improved consequence of cerebral malaria	↓ TNFα, IL-6, IL-1β Antimalarial activity	Val et al., 2015 [73]
AbM extract, p.o.	Rats, Pulmonary inflammation	↓ Lung damage induced by carcinogen	Attenuation of pulmonary inflammation & gross consolidation	Croccia et al., 2013 [74]
Erinacine A-enriched HE mycelia, p.o.	Mice, Life-prolonging activity	Increased longevity in aged mice	Induction of endogenous antioxidant enzymes	Li et al., 2019 [75]
HE Polysaccharide, p.o.	Mice, Colitis	Attenuation of colitis, reversing of gut dysbiosis	Downregulation of oxidative stress and inflamm.-related signaling pathways, Maintaining intestinal barrier	Ren et al., 2018 [76]
HE alcohol extract & polysacc., p.o.	Rats, IBD	Improved damages in colonic mucosa of induced IBD	↓ MPO activ., NF_K_B, TNFα, ↑T cc activ. Beneficial gut bacteria growth and improved host immunity	Diling et al., 2017 [77]
HE mycelium alcohol extract & erinacine A, p.o.	Rats, brain ischemia	Protection against brain ischemia injury induced neuronal cell death	Inhibition of iNOS/P3 MAPK, ↓ IL-1β, IL-6, TNFα, ↑ nerve growth properties	Lee et al., 2014 [78]
GF polysacc., p.o.	Rats, non-alcoholic steato-hepatitis	Protection	Beneficial regulation of microbiota	Li et al., 2019 [79]
Fermented GF extract, p.o.	Rats ET-induced uveitis	Anti-inflammatory	↓ IL-1β, TNFα, NF_Κ_B activ., iNOS express.	Han et al.., 2012 [80]
GF water extract, p.o.	Rats, IBD	↓ Colon ulceration	Amelioration by ↓MPO, TNFα colon express. and NF_Κ_B signaling	Lee et al., 2010 [81]

Abbreviations: tumor necrosis factor (TNF), inflammatory bowel disease (IBD), myeloperoxidase (MPO), nuclear factor kappa B (NF_K_B), inducible nitric oxide synthase (iNOS), mitogen-activated protein kinase (MAPK), endotoxin (ET).

**Table 5 nutrients-12-01339-t005:** Anti-inflammatory effect of *Agaricus blazei* Murill—Clinical studies.

Product, Admin.	Study In, Of	Effects	Mechanism	Author, Year [Ref]
AbM mycelia water extract incl. HE, GF, p.o. (Andosan)	Placebo-ctr RCT, IBD patients; 50 UC, 50 CD	Improved symptoms & QoL espec. in UC	↓ Proinflammatory effect	Therkelsen et al., 2016 [82,83,84]
AbM mycelia water extract incl. HE, GF, p.o. (Andosan)	Pilot study, IBD patients; 10 UC, 11 CD	Anti-inflammatory	↓ Proinflammatory cytokines, ↓ fecal calprotectin	Førland et al., 2011 [85]
AbM mycelia water extract incl. HE, GF, p.o. (Andosan)	Healthy Volunteers (*n* = 8)	Antioxidant effect	↓iROS prod. and Adhesion molec. express. in MΦ and granuloc.	Johnson et al., 2012 [86]
AbM mycelia water extract incl. HE, GF, p.o. (Andosan)	Healthy Volunteers (*n* = 10)	Predominantly anti-inflammatory effect	↓Proinflammatory cytokines	Johnson et al., 2009 [87]

Abbreviations: ulcerative colitis (UC), Crohn’s disease (CD), intracellular reactive oxygen species (iROS).

**Table 6 nutrients-12-01339-t006:** Anti-allergic Effects of *Agaricus blazei* Murill and *Grifola frondosa*
(Human study).

Product, Admin.	Study In, Of	Effects	Mechanism	Author, Year, [Ref.]
AbM mycelia water extract incl. HE, GF, p.o. (Andosan)	Placebo-ctr RCT in blood donors, Pollen allergy & asthma (*n* = 60)	↓ General symptoms, and medication	↓ Spec. IgE, reduced basophil sensitivity	Mahmood et al., 2019 [95]
Water AbM extract, p.o.	Mice, Allergy	↓ OVA sensitization	↓ Spec. IgE, improv. Th1/Th2 balance via MΦ activ. by epithelial cc, diff. promotion of naïve T cc to Th1 cc	Bouike et al., 2011 [94]
AbM mycelia water extract incl. HE, GF, p.o. (Andosan)	Mice, Allergy	↓ OVA sensitization	↓ Spec. IgE, IgG1 and improved Th1/Th2 balance	Ellertsen & Hetland 2009 [15]
AbM water extract, p.o.	Mice, Asthma	↓ Spec. IgE, IgG1 and bronchial eosinophils	Amelioration of skewed Th1/Th2 balance	Takimoto et al., 2008 [93]
AbM water extract, p.o.	Mice, Anaphylaxis	Inhib. of induced anaphylactic reaction and ear swelling	Treatment of mast cell mediated anaphylactic reaction	Choi et al., 2006 [92]
GF alcohol extract and ergosterol, p.o.	Mice, Allergic inflammation	Inhib. mast cc degranulation, alleviated anaphylactic cutaneous response	↓ Type 1 allergic reaction by suppression of mast cc degranulation	Kawai et al., 2019 [96]
GF polysacch, p.o.	Mice, AD	Inhib. AD-like skin lesion	↓ IgE, mast cc infiltr., cytokine express. controlling Th1/Th2	Park et al., 2015 [97]

Abbreviations: immunoglobulin (Ig), ovalbumin (OVA), atopic dermatitis (AD).

**Table 7 nutrients-12-01339-t007:** Safety of *Agaricus blazei* Murill, *Grifola frondosa* and *Hericium erinaceus*
(Human Studies).

Product Admin.	Study In, Of	Effects	Mechanism	Author, Year [Ref]
AbM powder, p.o.	Rats, Toxicity and oncogenicity	↓ Mortality in 2-yrs toxicity study	Possible anti-mutagenic and antioxidant effect, No carcinogenicity	Lee et al., 2008 [103]
AbM water extract, p.o.	Rats, Subchronic toxicity	Low subchronic toxicity at very high doses	Neg. genotoxicity test, possible clastogenic activity but no direct effect on DNA	Sumiya et al., 2008 [102]
AbM water extract, p.o.	Human case (*n* = 1), Contact dermatitis	Allergic chron. cheilitis, pos. delayed reaction after patch testing	Daily AbM intake for 6 months	Suehiro et al., 2007 [99]
AbM extract, p.o.	Cancer patients (*n* = 3), Liver function	Severe hepatic dysfunction	-	Mukai et al., 2006 [98]
GF polysacc., p.o.	Phase I/II safety study in 34 breast ca. pat.	No dose-limiting toxicity	-	Deng et al., 2009 [105]
GF spore, inhalation	Human case (*n* = 1), pneumonitis	Occupational hypersensitivity pneumonitis	Work for 3 mo. in mushroom farm	Tanaka et al., 2004 [104]
Erinacine A-enriched HE mycelia, p.o.	Mice, Life-prolonging activity	Increased longevity in aged mice	Induction of endogenous antioxidant enzymes	Li et al., 2019 [75]

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
