# Peer review of "Antitumor, Anti-inflammatory and Antiallergic Effects of Agaricus blazei Mushroom Extract and the Related Medicinal Basidiomycetes Mushrooms, Hericium erinaceus and Grifola frondosa: A Review of Preclinical and Clinical Studies"

_nutrients, 2020, doi:10.3390/nu12051339_

Round 1
Reviewer 1 Report
This article contains informative values as a review on medicinal use of Agaricus, although this topics have been discussed for certain extent elsewhere. However, the manuscript needs major reversion on the style and order of presentation.
This article contains informative values as a review on medicinal use of Agaricus, although this topics have been discussed for certain extent elsewhere. However, the manuscript needs major reversion on the style and order of presentation.
Authors need to show the reason why three mushrooms were picked up in this review, probably because they are the components of AndosanTM with that clinical studies are rather extended. This should be described in the introduction. Currently, several mushrooms in Basidiomycota family having similar functions are reported.
To make clear the author's intention to discuss the relationship of anti-tumor effects of Basidiomycota mushrooms with their anti-allergic and anti-inflammatory activities as described in the last sentences of introduction section, the preclinical studies should be discussed on the three mushrooms together, and also on the activities of interest, and then discuss the clinical studies of mushrooms together with Andosan.
Author Response
Reviewer 1:
Authors need to show the reason why three mushrooms were picked up in this review, probably because they are the components of AndosanTM with that clinical studies are rather extended. This should be described in the introduction. Currently, several mushrooms in Basidiomycota family having similar functions are reported.
This is now mentioned at end of 1st paragraph in Introduction.
To make clear the author's intention to discuss the relationship of anti-tumor effects of Basidiomycota mushrooms with their anti-allergic and anti-inflammatory activities as described in the last sentences of introduction section, the preclinical studies should be discussed on the three mushrooms together, and also on the activities of interest, and then discuss the clinical studies of mushrooms together with Andosan
In order to also agree with Reviewer 3, the tables are now divided into preclinical and clinical studies when appropriate (Table 1 and 3), but the separation into results with each mushroom is kept.
Reviewer 2 Report
This is an interesting manuscript which describe Antitumor, anti-inflammatory and antiallergic effects mushroom species. This work is important from the general perspective. Topic of the manuscript falls within the scope of the journal.
The title reflects the manuscript content. Abstract reflects manuscript content. All relevant parts of the manuscript are summarised. Keywords are well-chosen.
Introduction,
In the introduction, authors address medicinal propertis of mushrooms. However, I miss a few sentences on general describing of Agaricus blazei, Hericium erinaceus and Grifola frondosa (habitat, edibility ...)?
Conclusions are sound.
There are no conflicts of interests determined.
References
Quite a high number of references, which are current and representative.
Artwork: Tables are informative.
Author Response
Reviewer 2:
In the introduction, authors address medicinal propertis of mushrooms. However, I miss a few sentences on general describing of Agaricus blazei, Hericium erinaceus and Grifola frondosa (habitat, edibility ...)? This now further described in 1st sentence of Introduction, and in 2nd paragraph of Introduction.
Reviewer 3 Report
The review by Hetland et al., describes preclinical and clinical findings regarding the anticancer, anti-inflammatory, and antiallergic effects of three related Basidiomycota mushrooms AgaricusblazeiMurril, Hericium erinaceus and Grifolafrondosa with special focus on their antitumor effects, the mushrooms’ anti-inflammatory and antiallergic properties have also been described.
The paper falls within the scope of Nutrients. Major revision is requested in order to accept the manuscript. I hope that authors improve the points listed below.
The authors shall improve the following points:
- This review does not report “selection criteria”, used to select articles for Inclusion or exclusion criteria must be added in order to avoid any biases in the choice.
- Tables do not report information regarding the kind of trial, administration mode, and number of subjects.
- It is not clear if the main results reported by authors are due to the Beta-glucans fraction present in the extracts mentioned. The extraction procedure or the chemical composition of the described extracts has to be reported. Which kind of extract is Andosan?
- The second sentence in the discussion section is inappropriate, because erinaceus displays antitumor effects and G. frondosa displays anti-inflammatory properties too.
Complement chapter 2 with a new paragraph regarding the antitumor effects of HE (even just in vitro).Check for example:
- Lee SR, Jung K, Noh HJ, Park YJ, Lee HL, Lee KR, Kang KS, Kim KH. A new cerebroside from the fruiting bodies of Hericium erinaceus and its applicability to cancer treatment. Bioorg Med Chem Lett. 2015 Dec 15;25(24):5712-5. doi: 10.1016/j.bmcl.2015.10.092. Epub 2015 Nov 2.
- Kuo HC, Kuo YR, Lee KF, Hsieh MC, Huang CY, Hsieh YY, Lee KC, Kuo HL, Lee LY, Chen WP, Chen CC, Tung SY. A Comparative Proteomic Analysis of Erinacine A's Inhibition of Gastric Cancer Cell Viability and Invasiveness. Cell PhysiolBiochem. 2017;43(1):195-208. doi: 10.1159/000480338. Epub 2017 Aug 30.
- Wang M, Zhang Y, Xiao X, Xu D, Gao Y, Gao Q. A Polysaccharide Isolated from Mycelia of the Lion's Mane Medicinal Mushroom Hericium erinaceus (Agaricomycetes) Induced Apoptosis in Precancerous Human Gastric Cells. Int J Med Mushrooms. 2017;19(12):1053-1060. doi: 10.1615/IntJMedMushrooms.2017024975.
- Kim SP, Nam SH, Friedman M. Hericium erinaceus (Lion's Mane) mushroom extracts inhibit metastasis of cancer cells to the lung in CT-26 colon cancer-tansplanted mice. J Agric Food Chem. 2013 May 22;61(20):4898-904. doi: 10.1021/jf400916c. Epub 2013 May 13
- Li G, Yu K, Li F, Xu K, Li J, He S, Cao S, Tan G. Anticancer potential of Hericium erinaceus extracts against human gastrointestinal cancers. J Ethnopharmacol. 2014 Apr 28;153(2):521-30. doi: 10.1016/j.jep.2014.03.003. Epub 2014 Mar 12.
- Zan X, Cui F, Li Y, Yang Y, Wu D, Sun W, Ping L. Hericium erinaceus polysaccharide-protein HEG-5 inhibits SGC-7901 cell growth via cell cycle arrest and apoptosis. Int J BiolMacromol. 2015 May;76:242-53. doi: 10.1016/j.ijbiomac.2015.01.060. Epub 2015 Feb 20.
- Complement chapter 3 with a new paragraph regarding the anti-inflammatory effects of GF.Check for example:
- Chien RC, Yang YC, Lai EI, Mau JL. Anti-Inflammation and Lipogenic Inhibition of Taiwanofungussalmonea Mycelium and Grifolafrondosa Fruiting Body. Int J Med Mushrooms. 2017;19(7):629-640. doi: 10.1615/IntJMedMushrooms.2017021239.
- Chien RC, Lin LM, Chang YH, Lin YC, Wu PH, Asatiani MD, Wasser SG, Krakhmalnyi M, Agbarya A, Wasser SP, Mau JL. Anti-Inflammation Properties of Fruiting Bodies and Submerged Cultured Mycelia of Culinary-Medicinal Higher Basidiomycetes Mushrooms. Int J Med Mushrooms. 2016;18(11):999-1009. doi: 10.1615/IntJMedMushrooms.v18.i11.50.
- Li X, Zeng F, Huang Y, Liu B. The Positive Effects of GrifolafrondosaHeteropolysaccharide on NAFLD and Regulation of the Gut Microbiota. Int J Mol Sci. 2019 Oct 24;20(21). pii: E5302. doi: 10.3390/ijms20215302.
- Chien RC, Lin LM, Chang YH, Lin YC, Wu PH, Asatiani MD, Wasser SG, Krakhmalnyi M, Agbarya A, Wasser SP, Mau JL. Anti-Inflammation Properties of Fruiting Bodies and Submerged Cultured Mycelia of Culinary-Medicinal Higher Basidiomycetes Mushrooms. Int J Med Mushrooms. 2016;18(11):999-1009. doi: 10.1615/IntJMedMushrooms.v18.i11.50.
- Wu SJ, Lu TM, Lai MN, Ng LT. Immunomodulatory activities of medicinal mushroom Grifolafrondosa extract and its bioactive constituent.Am J Chin Med. 2013;41(1):131-44. doi: 10.1142/S0192415X13500109.
- Rossi P, Difrancia R, Quagliariello V, Savino E, Tralongo P, Randazzo CL, Berretta M. B-glucans from Grifolafrondosa and Ganodermalucidum in breast cancer: an example of complementary and integrative medicine. Oncotarget. 2018 May 15;9(37):24837-24856. doi: 10.18632/oncotarget.24984. eCollection 2018 May 15. Review.
- Han C, Cui B. Pharmacological and pharmacokinetic studies with agaricoglycerides, extracted from Grifolafrondosa, in animal models of pain and inflammation. 2012 Aug;35(4):1269-75. doi: 10.1007/s10753-012-9438-5.
- Lee JS, Park SY, Thapa D, Choi MK, Chung IM, Park YJ, Yong CS, Choi HG, Kim JA. Grifolafrondosa water extract alleviates intestinal inflammation by suppressing TNF-alpha production and its signaling. ExpMol Med. 2010 Feb 28;42(2):143-54. doi: 10.3858/emm.2010.42.2.016.
- The contradictory proinflammatory or anti-inflammatory effect of Andosan is only speculative and I would like to remind the authors the paper describing a dose dependent effect of citokines levels of GF extract in breast cancer patients (Deng et al. 2009).
Minor comments
- It could be better to divide the Tables on preclinical and clinical studies, following the logical presentation of the manuscript.
- Add notes for figure 1.
- Describe the percentage of ingredients of HE and GF in the Andosan supplement.
- Remove any consideration about a possible change in microbiota composition. The paper does not address this matter.
Author Response
Reviewer 3:
The authors shall improve the following points:
- This review does not report “selection criteria”, used to select articles for Inclusion or exclusion criteria must be added in order to avoid any biases in the choice. Now in Introduction after aim.
- Tables do not report information regarding the kind of trial, administration mode, and number of subjects. Is now included.
- It is not clear if the main results reported by authors are due to the Beta-glucans fraction present in the extracts mentioned. This is still unknown, as written at end of Discussion, although they are thought to be due to local stimulation by beta-glucans of GALT together with systemic action of absorbable less defined LMW substances .The extraction procedure or the chemical composition of the described extracts has to be reported. This is now done in the Tables.
Which kind of extract is Andosan? It is a water extract of AbM, HE (15%) and GF (3%) mycelium, as stated in text and Tables. - The second sentence in the discussion section is inappropriate, because erinaceus displays antitumor effects and G. frondosa displays anti-inflammatory properties too. It is now changed accordingly.
- Complement chapter 2 with a new paragraph regarding the antitumor effects
of HE (even just in vitro). Check for example:
- Lee SR, Jung K, Noh HJ, Park YJ, Lee HL, Lee KR, Kang KS, Kim KH. A new cerebroside from the fruiting bodies of Hericium erinaceus and its applicability to cancer treatment. Bioorg Med Chem Lett. 2015 Dec 15;25(24):5712-5. doi: 10.1016/j.bmcl.2015.10.092. Epub 2015 Nov 2.
- Kuo HC, Kuo YR, Lee KF, Hsieh MC, Huang CY, Hsieh YY, Lee KC, Kuo HL, Lee LY, Chen WP, Chen CC, Tung SY. A Comparative Proteomic Analysis of Erinacine A's Inhibition of Gastric Cancer Cell Viability and Invasiveness. Cell PhysiolBiochem. 2017;43(1):195-208. doi: 10.1159/000480338. Epub 2017 Aug 30.
- Wang M, Zhang Y, Xiao X, Xu D, Gao Y, Gao Q. A Polysaccharide Isolated from Mycelia of the Lion's Mane Medicinal Mushroom Hericium erinaceus (Agaricomycetes) Induced Apoptosis in Precancerous Human Gastric Cells. Int J Med Mushrooms. 2017;19(12):1053-1060. doi: 10.1615/IntJMedMushrooms.2017024975.
- Kim SP, Nam SH, Friedman M. Hericium erinaceus (Lion's Mane) mushroom extracts inhibit metastasis of cancer cells to the lung in CT-26 colon cancer-tansplanted mice. J Agric Food Chem. 2013 May 22;61(20):4898-904. doi: 10.1021/jf400916c. Epub 2013 May 13
- Li G, Yu K, Li F, Xu K, Li J, He S, Cao S, Tan G. Anticancer potential of Hericium erinaceus extracts against human gastrointestinal cancers. J Ethnopharmacol. 2014 Apr 28;153(2):521-30. doi: 10.1016/j.jep.2014.03.003. Epub 2014 Mar 12.
- Zan X, Cui F, Li Y, Yang Y, Wu D, Sun W, Ping L. Hericium erinaceus polysaccharide-protein HEG-5 inhibits SGC-7901 cell growth via cell cycle arrest and apoptosis. Int J BiolMacromol. 2015 May;76:242-53. doi: 10.1016/j.ijbiomac.2015.01.060. Epub 2015 Feb 20.There is now a new chapter on p.8: «2.d. Antitumor effects of HE« with new refs # 69-71 (Kim et al., 2011; Kim et al., 2013; Li et al., 2014), on in vivo antitumor effects of HE.
- Complement chapter 3 with a new paragraph regarding the anti-inflammatory effects of GF. Check for example:
- Chien RC, Yang YC, Lai EI, Mau JL. Anti-Inflammation and Lipogenic Inhibition of Taiwanofungussalmonea Mycelium and Grifolafrondosa Fruiting Body. Int J Med Mushrooms. 2017;19(7):629-640. doi: 10.1615/IntJMedMushrooms.2017021239.
- Chien RC, Lin LM, Chang YH, Lin YC, Wu PH, Asatiani MD, Wasser SG, Krakhmalnyi M, Agbarya A, Wasser SP, Mau JL. Anti-Inflammation Properties of Fruiting Bodies and Submerged Cultured Mycelia of Culinary-Medicinal Higher Basidiomycetes Mushrooms. Int J Med Mushrooms. 2016;18(11):999-1009. doi: 10.1615/IntJMedMushrooms.v18.i11.50.
- Li X, Zeng F, Huang Y, Liu B. The Positive Effects of GrifolafrondosaHeteropolysaccharide on NAFLD and Regulation of the Gut Microbiota. Int J Mol Sci. 2019 Oct 24;20(21). pii: E5302. doi: 10.3390/ijms20215302.
- Chien RC, Lin LM, Chang YH, Lin YC, Wu PH, Asatiani MD, Wasser SG, Krakhmalnyi M, Agbarya A, Wasser SP, Mau JL. Anti-Inflammation Properties of Fruiting Bodies and Submerged Cultured Mycelia of Culinary-Medicinal Higher Basidiomycetes Mushrooms. Int J Med Mushrooms. 2016;18(11):999-1009. doi: 10.1615/IntJMedMushrooms.v18.i11.50.
- Wu SJ, Lu TM, Lai MN, Ng LT. Immunomodulatory activities of medicinal mushroom Grifolafrondosa extract and its bioactive constituent.Am J Chin Med. 2013;41(1):131-44. doi: 10.1142/S0192415X13500109.
- Rossi P, Difrancia R, Quagliariello V, Savino E, Tralongo P, Randazzo CL, Berretta M. B-glucans from Grifolafrondosa and Ganodermalucidum in breast cancer: an example of complementary and integrative medicine. Oncotarget. 2018 May 15;9(37):24837-24856. doi: 10.18632/oncotarget.24984. eCollection 2018 May 15. Review.
- Han C, Cui B. Pharmacological and pharmacokinetic studies with agaricoglycerides, extracted from Grifolafrondosa, in animal models of pain and inflammation. 2012 Aug;35(4):1269-75. doi: 10.1007/s10753-012-9438-5.
- Lee JS, Park SY, Thapa D, Choi MK, Chung IM, Park YJ, Yong CS, Choi HG, Kim JA. Grifolafrondosa water extract alleviates intestinal inflammation by suppressing TNF-alpha production and its signaling. ExpMol Med. 2010 Feb 28;42(2):143-54. doi: 10.3858/emm.2010.42.2.016.
- The contradictory proinflammatory or anti-inflammatory effect of Andosan is only speculative and I would like to remind the authors the paper describing a dose dependent effect of citokines levels of GF extract in breast cancer patients (Deng et al. 2009).
- Minor comments
- There is a new paragraph on Anti-inflammatory properties of GF, end of p. 9: «3.c. GF», and with incorporation of 3 preclinical GF studies, new refs # 89, 90, 91 (Li et al., 2019; Han et al. 2012; Lee et al., 2010) inTable 3 a.
- It could be better to divide the Tables on preclinical and clinical studies, following the logical presentation of the manuscript. This is done for Table 1 and 3, which is now a) and b), but not for Tables 2 and 4 where there was only one clinical study in each.
- Add notes for figure 1. Notes were originally added for Figure 1 above the figure on p.6 and also in Figure 1 legend and the 3rd last paragraph of Discussion.
- Describe the percentage of ingredients of HE and GF in the Andosan supplement. This is now done in Introduction.
- Remove any consideration about a possible change in microbiota composition. The paper does not address this matter. This now done: The 4th paragraph of Discussion mentioning microbiota, is now removed together with former refs # 99 and 100.
- About references: New refs are: # 11,12,42, 58, 69, 70, 71,89, 90, 91, 109, and 111.Ref 54: 1 author is Xiaohui T.
- Ref 104 by Tanaka et al 2004 was wrong – is now rectified
- Ref 41 by Murakawa et al 2007 was wrong – is now rectified
- Previous refs 1 and 2 have changed place.
Round 2
Reviewer 1 Report
Clinical studies on Andosan are mainly provided but more extended discussion needs to rationalize the antit cancer effect based on such as synergy effect among the three mushrooms.
Reviewer 3 Report
The manuscript has been improved as suggested. Any additional comments. I recommend this paper for acceptance